# New Insights into the Lactic Acid Resistance Determinants of *Listeria monocytogenes* Based on Transposon Sequencing and Transcriptome Sequencing Analyses

Xiayu Liu,[b] Xinxin Pang,[b] Yansha Wu,[b] Yajing Wu,[b] Linan Xu,[a] ⓘ Qihe Chen,[b] Jianrui Niu,[a] ⓘ Xinglin Zhang[a,b]

[a]College of Agriculture and Forestry, Linyi University, Linyi, China
[b]Department of Food Science and Nutrition, Zhejiang University, Hangzhou, China

Xiayu Liu and Xinxin Pang contributed equally to this work. Author order was determined both alphabetically and in order of increasing seniority.

**ABSTRACT** *Listeria monocytogenes* is a foodborne pathogen that can tolerate a variety of extreme environments. In particular, its acid resistance (AR) capability is considered one of the key factors threating food safety. Here, we employed a microbial functional genomic technology termed transposon sequencing (Tn-seq), leading to the identification of two genes involved in cell wall peptidoglycan biosynthesis (*murF*) and phosphate transport (*lmo2248*) that play key roles in lactic acid resistance (LAR) of *L. monocytogenes*. Deletion of *lmo2248* significantly impaired the ability of LAR in *L. monocytogenes*, demonstrating the accuracy of the Tn-seq results. Transcriptome analysis revealed that 31.7% of the *L. monocytogenes* genes on the genome were differentially expressed under lactic acid (LA) treatment, in which genes involved in phosphate transport were influenced most significantly. These findings shed light on the LAR mechanisms of *L. monocytogenes*, which may contribute to the development of novel strategies against foodborne pathogens.

**IMPORTANCE** *Listeria monocytogenes* is a Gram-positive foodborne pathogen with high lethality and strong stress resistance, and its strong acid tolerance leads to many foodborne illnesses occurring in low-pH foods. Lactic acid is a generally recognized as safe (GRAS) food additive approved for use by the FDA. However, the genetic determinants of lactic acid resistance in *L. monocytogenes* have not been fully identified. In this study, the lactic acid resistance determinants of *L. monocytogenes* were comprehensively identified by Tn-seq on a genome-wide scale. Two genes, *murF* (cell wall peptidoglycan biosynthesis) and *lmo2248* (phosphate transport), were identified to play an important role in the lactic acid resistance. Moreover, genome-wide transcriptomic analysis showed that phosphotransferase system (PTS)-related genes play a key role at the transcriptional level. These findings contribute to a better understanding of the lactic acid resistance mechanism of *L. monocytogenes* and may provide unique targets for the development of other novel antimicrobial agents.

**KEYWORDS** lactic acid, *Listeria monocytogenes*, RNA-seq, Tn-seq

*Listeria monocytogenes* is a common Gram-positive, facultative, anaerobic bacterium and one of the most fatal foodborne pathogens worldwide (1). It is mainly transmitted by food and widely found in meat, eggs, poultry, seafood, dairy products, and vegetables (2). In immunocompromised individuals, even low levels of food contamination (approximately $10^2$ to $10^4$ bacteria) can cause symptoms such as bacterial sepsis, meningitis, and abortion to pregnancy, with mortality rates as high as 20 to 30% (3). Foodborne diseases caused by *L. monocytogenes* infection have remained a major problem in the food industry over the past decade (4, 5).

In recent years, the rapid emergence of antibiotic-resistant bacteria has gradually reduced the effectiveness of antibiotics (6). Therefore, considerable effort has been

Address correspondence to Xinglin Zhang, zhangxinglin@lyu.edu.cn, or Jianrui Niu, niujianrui@lyu.edu.cn.

The authors declare no conflict of interest.

10.1128/spectrum.02750-22 **1**

devoted to finding some efficient natural antimicrobials (7, 8). Organic acids (lactic acid, malic acid, citric acid, propionic acid, acetic acid, etc.) are widely used as preservatives and antimicrobial agents in meat products, dairy products, and other foods because of their advantages of low price, simple processing, and significant bacteriostatic effect (9). As a representative organic acid, lactic acid has been widely reported to exhibit broad-spectrum antibacterial activity against Gram-negative and Gram-positive pathogens. Wang et al. (10) found that 0.5% lactic acid could completely inhibit the growth of *Salmonella enterica* serovar Enteritidis, *Escherichia coli*, and *L. monocytogenes*, while causing a large amount of protein leakage. Moreover, both the FDA and the European Food Safety Authority have approved the addition of lactic acid (LA) for the food industry as an antibacterial agent (11, 12). However, many bacteria have been reported to have the ability to adapt to acid environments and survive under low-pH conditions. For instance, Yu et al. (13) found that *E. coli* O157 and O26 could survive for more than 12 h after exposure to pH 3.5. Mani-López et al. found (9, 14) that *Salmonella* is particularly capable of adapting to acidic environments and surviving under intense pH conditions and that its inducible acid tolerance response (ATR) is critical for its low pH tolerance. As a result, many foodborne illness outbreaks occur in low-pH foods (yogurt, juice, cheese, etc.). Hence, it is important to study the acid resistance (AR) mechanisms of foodborne pathogens. Previous studies have shown that the adaptive ATR mechanism of *L. monocytogenes* is closely related to the LisRK 2-component regulatory system, the SOS response, components of the $\sigma^B$ regulon, changes in membrane fluidity, the F0F1-ATPase proton pump, the glutamate decarboxylase (GAD) system, and the arginine deiminase (ADI) system (15, 16). Moreover, Begley et al. and Madeo et al. also found that genes *btlA* and *thiT* contribute to the acid tolerance of *L. monocytogenes* (17, 18). However, many LA resistance (LAR)-related genes have not yet been fully identified, so we need to further explore the LAR mechanism of *L. monocytogenes* from a genome-wide level.

Transposon sequencing (Tn-seq) is a powerful method for the study of bacterial functional genomics, which combines transposon mutagenesis with high-throughput sequencing (19). It can directly link phenotype to genotype in a genome-wide scale (20, 21). Tn-seq technology facilities comprehensive identification of conditionally essential genes of foodborne pathogens in particular growth environments. For instance, Zhang et al. (22) have identified 37 genes that are essential for the growth of *Enterococcus faecium* in human serum. Jayeola et al. (23) have revealed the essential genes that contribute to the growth of *Cronobacter sakazakii* and *Salmonella* in low-moisture foods (LMFs). In addition, Tn-seq also plays an important role in the identification of antibiotic resistance genes in many clinically important pathogens (including the ESKAPE [*E. faecium*, *Staphylococcus aureus*, *Klebsiella pneumoniae*, *Acinetobacter baumannii*, *Pseudomonas aeruginosa*, and *Enterobacter*] species) (24–26). In this study, Tn-seq and transcriptome sequencing (RNA-seq) were employed to investigate the lactic acid resistance (LAR) mechanisms in *L. monocytogenes*, which may lead to a deeper understanding of the LAR mechanism of *L. monocytogenes* in response to extreme pH and shed light on the development of novel antibacterial drugs.

## RESULTS

**Antibacterial effect of LA against *L. monocytogenes*.** As shown in Fig. 1, LA exhibited a significant inhibitory effect on the growth of *L. monocytogenes* and showed a significant dose-dependent manner. When the concentration of LA reached 4 mg/mL (pH 4.68), the growth of *L. monocytogenes* was completely inhibited. At 12 h, the addition of LA (1 mg/mL [pH 6.52] and 2 mg/mL [pH 5.92]) inhibited the growth of *L. monocytogenes* by 15% and 36%, respectively, compared with the control group. The result indicated that high doses of LA (≥4 mg/mL) were effective inhibitors for controlling *L. monocytogenes*, while *L. monocytogenes* showed significant acid resistance (AR) to low doses of LA (≤2 mg/mL).

**Identification of genetic determinants involved in LAR in *L. monocytogenes* by Tn-seq.** To identify the genes required for LAR in *L. monocytogenes*, Tn-seq was performed to determine which mutants were selectively lost during culturing in the

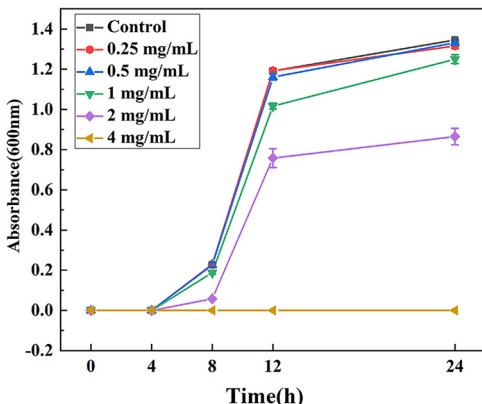

**FIG 1** Growth curves of *L. monocytogenes* under different concentrations of LA treatment. Data are presented as the mean ± standard deviation (*n* = 3).

presence of LA. Though 10 genes (fold change [FC], >2 and *P* < 0.05) were identified to be involved in LAR, only 2 genes (*murF* and *lmo2248*) had strict statistical significance (Benjamini-Hochberg corrected *P* value (BH) < 0.05) (Fig. 2 and Table 1). A single gene, *murF* (locus tag lmo0856), encoding a UDP-*N*-acetylmuramoylalanyl-D-glutamyl-2,6-diaminopimelate-D-alanyl-D-alanyl ligase, was identified by Tn-seq with the highest fold change (FC = 13.5). Another significant gene, *lmo2248* (FC, 4.4), was a putative phosphate transport regulator.

**Effect of LA on growth of *L. monocytogenes* wild type and Δ*lmo2248* mutant.** In order to validate the results of the Tn-seq data and to determine the role of the identified genes in LA resistance, we constructed a markerless deletion mutant, Δ*lmo2248*, in *L. monocytogenes*. Since *murF* is an essential gene (27), we selected the second-ranked gene, *lmo2248*, for further validation and functional studies. In the absence of LA, the growth of the wild-type strain was not significantly different from that of the mutant Δ*lmo2248* and the complemented strain Δ*lmo2248*+*lmo2248* (Fig. 3A). However, when the WT and Δ*lmo2248* were grown in brain heart infusion (BHI) with 1 mg/mL LA, the growth rate of the Δ*lmo2248* mutant decreased dramatically compared to that of the WT (Fig. 3B). Moreover, it can be visualized in Fig. 3C that the deletion strain of *lmo2248* was almost completely unable to grow on agar plates spiked with LA (2 mg/mL). The complemented strain Δ*lmo2248*+*lmo2248* could partially restore the LA resistance to the wild-type level (Fig. 3B and C). These findings suggest that gene *lmo2248* indeed contributes to LA resistance in *L. monocytogenes*.

**Morphological observation of *L. monocytogenes* and Δ*lmo2248*.** Scanning electron microscopy (SEM) and transmission electron microscopy (TEM) were applied to observe the effect of LA on the cell ultrastructure of *L. monocytogenes* (WT) and Δ*lmo2248*. As shown in Fig. 3D, under treatment with the same concentration of LA, obvious shrinkage and damage (red arrows) appeared on the cell surface of the Δ*lmo2248* mutant, while no significant changes appeared in the wild type. Similarly, in the internal morphological changes of the cells (Fig. 3E), the intracellular substances (nucleic acids and proteins) of the Δ*lmo2248* mutant were leaked in large quantities, and the structure of the cell membrane was severely damaged. Overall, these findings suggest that the Δ*lmo2248* mutant is more sensitive to LA-mediated bactericidal effects than the wild type. Therefore, the accuracy of the Tn-seq data was further validated, indicating that the *lmo2248* gene is indeed critical for LA resistance in *L. monocytogenes*.

**Transcriptomic analysis of *L. monocytogenes* during growth in rich medium and in LA.** Transcriptomic analysis was performed to analyze differential gene expression of *L. monocytogenes* under LA treatment. Compared with the control, 944 differentially expressed genes (DEGs) were identified in the LA-treated group, including 413 upregulated genes and 531 downregulated genes (Fig. 4A). Among these DEGs, *lmo2248* was also significantly upregulated (FC, 2.65; *P* < 0.01) in transcriptomic

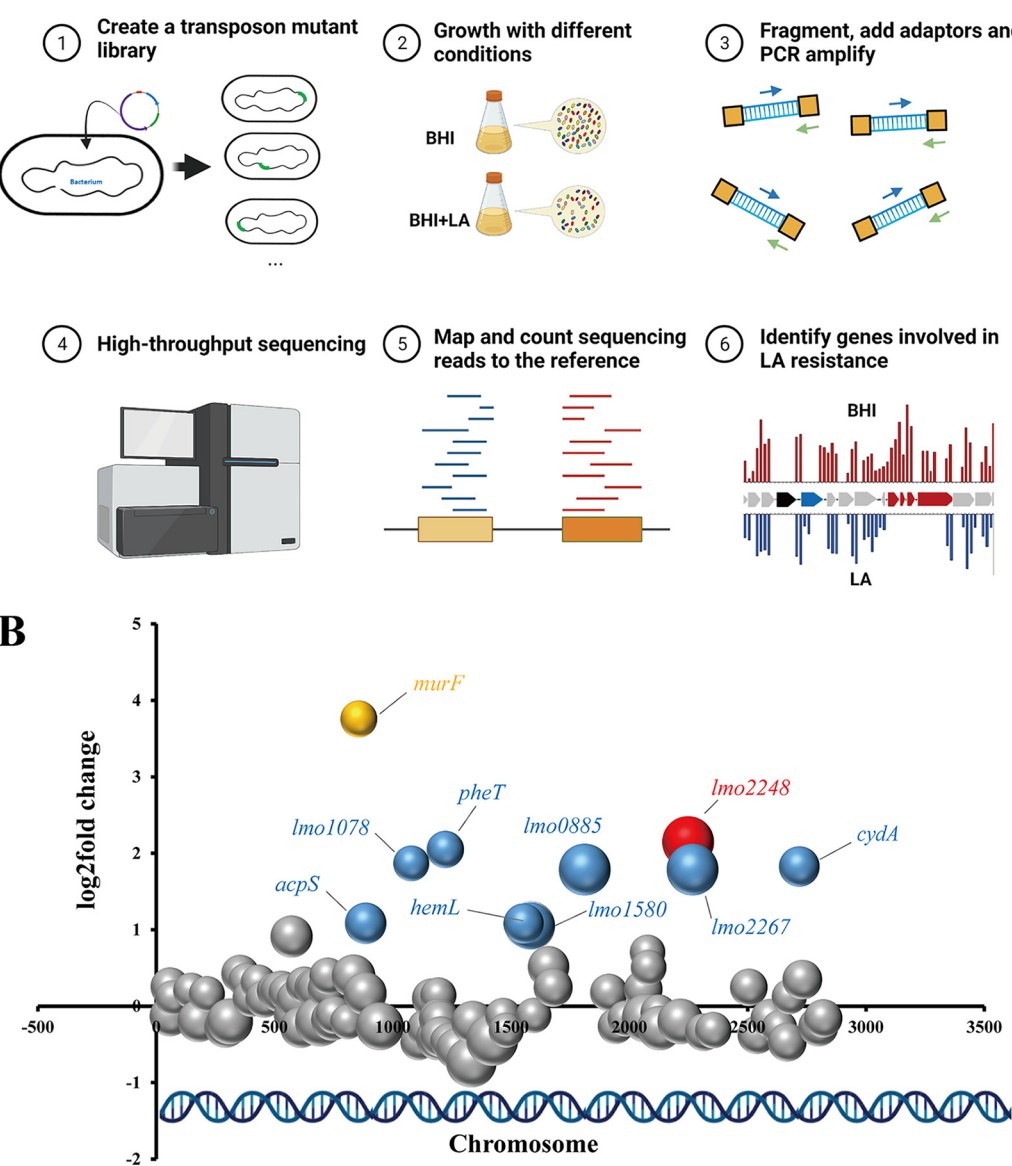

**FIG 2** Tn-seq analysis to identify *L. monocytogenes* genes involved in LA resistance. (A) Schematic depiction of the Tn-seq. (B) Identification of genes involved in LAR by Tn-seq analysis. Different bubbles represent different genes, and bubble size corresponds to different statistical analysis values (–log[*P*-values]). The larger the bubble size, the stronger the significance. The *x* axis represents the location of the gene in the chromosome, and the *y* axis represents the fold change value (LA treatment versus control). Genes with significant changes (*P* < 0.05) are represented by colored bubbles; other genes are represented in gray. Two genes with a BH of <0.05 are shown in yellow and red.

analysis. As can be seen from the heat map, the transcriptome profiles of the two groups were significantly different (Fig. 4B).

Gene Ontology (GO) enrichment analysis and KEGG pathway analysis were used to further analyze the function of DEGs. The most significant top 20 GO terms are shown in Fig. 4C. Upregulated genes were concentrated in the cellular component, while downregulated genes were concentrated in the biological process and molecular function categories. For biological process, the most notable subcategory is "phosphoenol-pyruvate-dependent sugar phosphotransferase system," and for cellular component and molecular function, "ribosome" and "protein-N(PI)-phosphohistidine-sugar phos-photransferase activity" were the most significant subcategories, respectively.

**TABLE 1** *L. monocytogenes* genes involved in lactic acid resistance as determined by Tn-seq analysis

| Synonym | Gene name | Annotation | Fold change[a] | P value | BH value[b] |
|---|---|---|---|---|---|
| *lmo0856* | *murF* | UDP-*N*-acetylmuramoylalanyl-D-glutamyl-2,6-diamino pimelate-D-alanyl-D-alanyl ligase | 13.5 | 3.6E-02 | 6.6E-03 |
| *lmo2248* | | hypothetical protein | 4.4 | 1.3E-03 | 6.6E-03 |
| *lmo1222* | *pheT* | Phenylalanyl-tRNA synthetase subunit beta | 4.1 | 3.3E-02 | NS |
| *lmo1078* | | UDP-glucose pyrophosphorylase | 3.7 | 4.2E-02 | NS |
| *lmo2718* | *cydA* | Cytochrome *d* ubiquinol oxidase subunit I | 3.6 | 1.7E-02 | NS |
| *lmo1811* | | ATP-dependent DNA helicase RecG | 3.5 | 1.3E-03 | NS |
| *lmo2267* | | ATP-dependent deoxyribonuclease subunit A | 3.5 | 1.3E-03 | NS |
| *lmo0885* | *acpS* | 4′-Phosphopantetheinyl transferase | 2.1 | 1.5E-02 | NS |
| *lmo1553* | *hemL* | Glutamate-1-semialdehyde aminotransferase | 2.1 | 1.9E-02 | NS |
| *lmo1580* | | Hypothetical protein | 2.1 | 2.1E-03 | NS |

[a]Indicates the fold change derived from the ratio of the unselected control library to the lactic acid competitively selected library.
[b]NS, not significant (BH value of >0.05).

KEGG pathway analysis of DEGs in *L. monocytogenes* was conducted in the LA treatment group compared to the control group. "Ribosome," "arginine biosynthesis," and "alanine, aspartate and glutamate metabolism" were the three most significantly upregulated pathways (Fig. 5A). The most significantly downregulated pathways included "phosphotransferase system (PTS)," "fructose and mannose metabolism," and "starch and sucrose metabolism" (Fig. 5B). In addition, the associations between different pathways were also shown in Fig. 5C. More interestingly, phosphate-related genes (Table 2) were significantly enriched in both GO and KEGG analyses, indicating that they play a crucial role in LA resistance.

**Validation of RNA-seq data by qPCR.** The accuracy of the transcriptome data was verified by real-time quantitative PCR (qPCR). As shown in Fig. 6, the RNA-seq and qPCR data of the LA-treated group and the control group were highly similar, which demonstrated the reliability and stability of the transcriptome analysis results.

## DISCUSSION

The addition of lactic acid can significantly inhibit the growth of *L. monocytogenes*, so it is widely used in the prevention and control of *L. monocytogenes* in the food industry (28, 29). However, the application and development of LA are often limited due to the strong acid resistance of *L. monocytogenes*. Therefore, it is very important to explore the lactic acid resistance (LAR) mechanism of *L. monocytogenes*.

The lack of suitable genetic tools has long been a bottleneck in the study of *L. monocytogenes* acid resistance mechanisms. In this study, we employed Tn-seq (19), a bacterial functional genomics research method that combines genome-wide transposon mutagenesis with high-throughput sequencing, to explore the LAR mechanism of *L. monocytogenes*.

Two novel LAR genes (*murF* and *lmo2248*) were identified through this genome-wide technology. Gene *murF* is mainly involved in the synthesis of cell wall peptidoglycan by catalyzing the final step in the synthesis of UDP-*N*-acetylmuramyl pentapeptide (the precursor of murein) (30). In *S. aureus*, the *murF* gene not only plays an important role in peptidoglycan biosynthesis, but may also be involved in the control of cell division (27). Gene *lmo2248* was annotated as a hypothetical protein; however, based on previous studies (31), it may be a putative regulator of phosphate transport. In addition, *lmo2249*, a downstream gene of *lmo2248*, was identified to encode a low-affinity inorganic phosphate transporter mainly responsible for phosphate transport (32). Hence, the gene *lmo2248* may promote LAR in *L. monocytogenes* by playing a role in phosphate transport. In addition, the significant upregulation of *lmo2248* in the transcriptomic analysis indicated that it also plays an important role in the LAR of *L. monocytogenes* at the transcriptional level.

The significant differences in growth and killing curves of markerless deletion mutant Δ*lmo2248* and the wild type in the presence of LA strongly illustrate the accuracy of the Tn-seq data and the importance of the identified genes in LAR. In addition, their

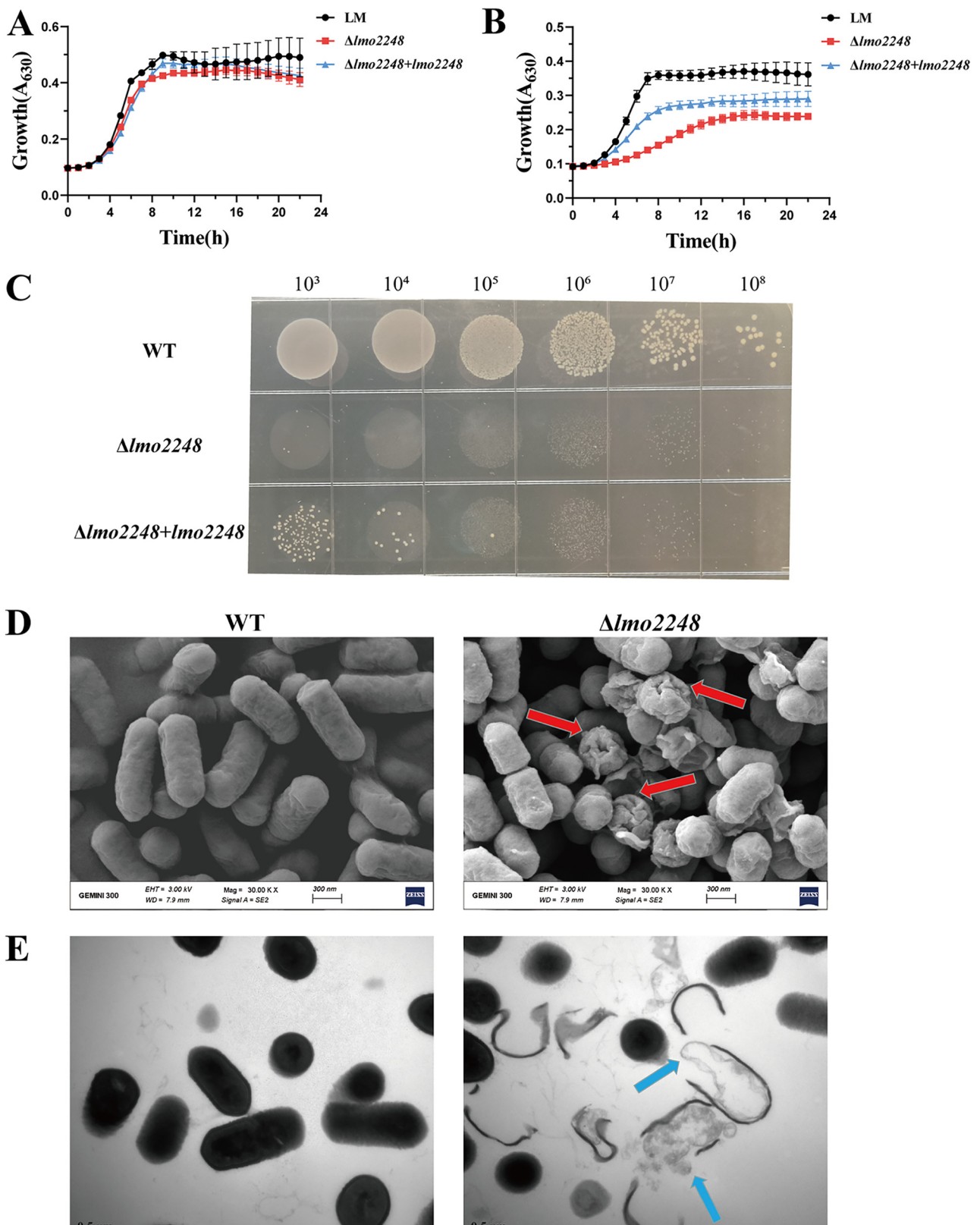

FIG 3 The effect of targeted mutation of *lmo2248* on growth and morphology of *L. monocytogenes* in the presence of LA. (A and B) Growth curve of wild-type *L. monocytogenes* (WT), Δ*lmo2248*, and Δ*lmo2248+lmo2248* in the absence (A) and presence (B) (1 mg/mL) of LA. (C) Growth of wild-type (WT), Δ*lmo2248*, and Δ*lmo2248+lmo2248* strains on agar plates. (D and E) SEM (D) and TEM (E) images of *L. monocytogenes* and Δ*lmo2248*. The WT and Δ*lmo2248* were both treated with LA (4 mg/mL) for 2 h.

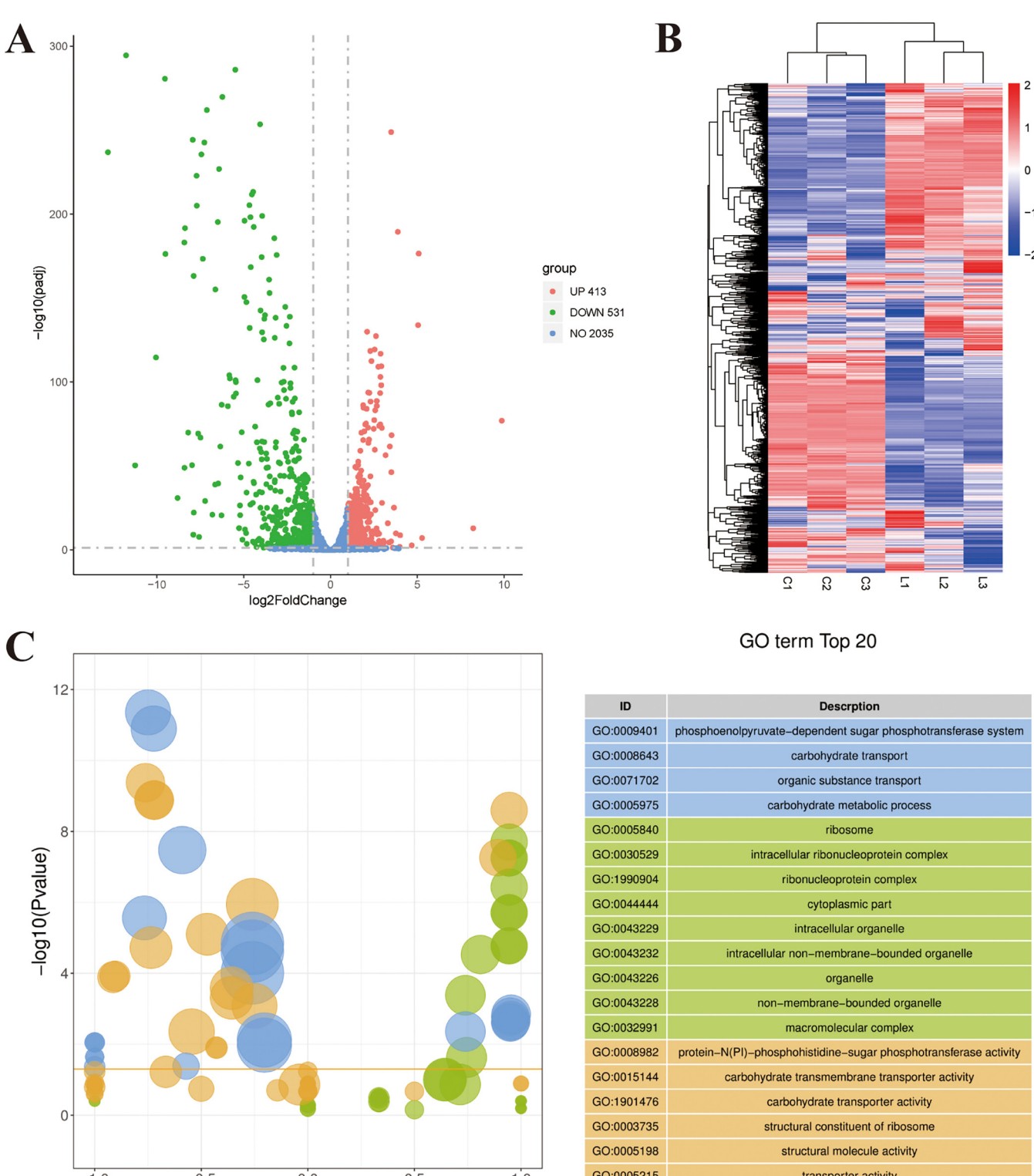

**FIG 4** Transcriptomic analysis of *L. monocytogenes* during growth in rich medium and in LA. (A) Volcano plot of the DEGs. (B) Heat map of DEGs. (C) Z-score bubble plot for GO enrichment analysis. The ordinate is $-\log_{10}(P$ value), the abscissa is the up-down normalization value (the ratio of the difference between the number of differentially upregulated genes and the number of differentially downregulated genes to the total differential genes), and the size of the bubble represents the target gene enriched by the current GO term number; the yellow line represents the threshold of a $P$ value of 0.05; the right side is the list of the top 20 terms of the $P$ value, and different colors represent different ontologies.

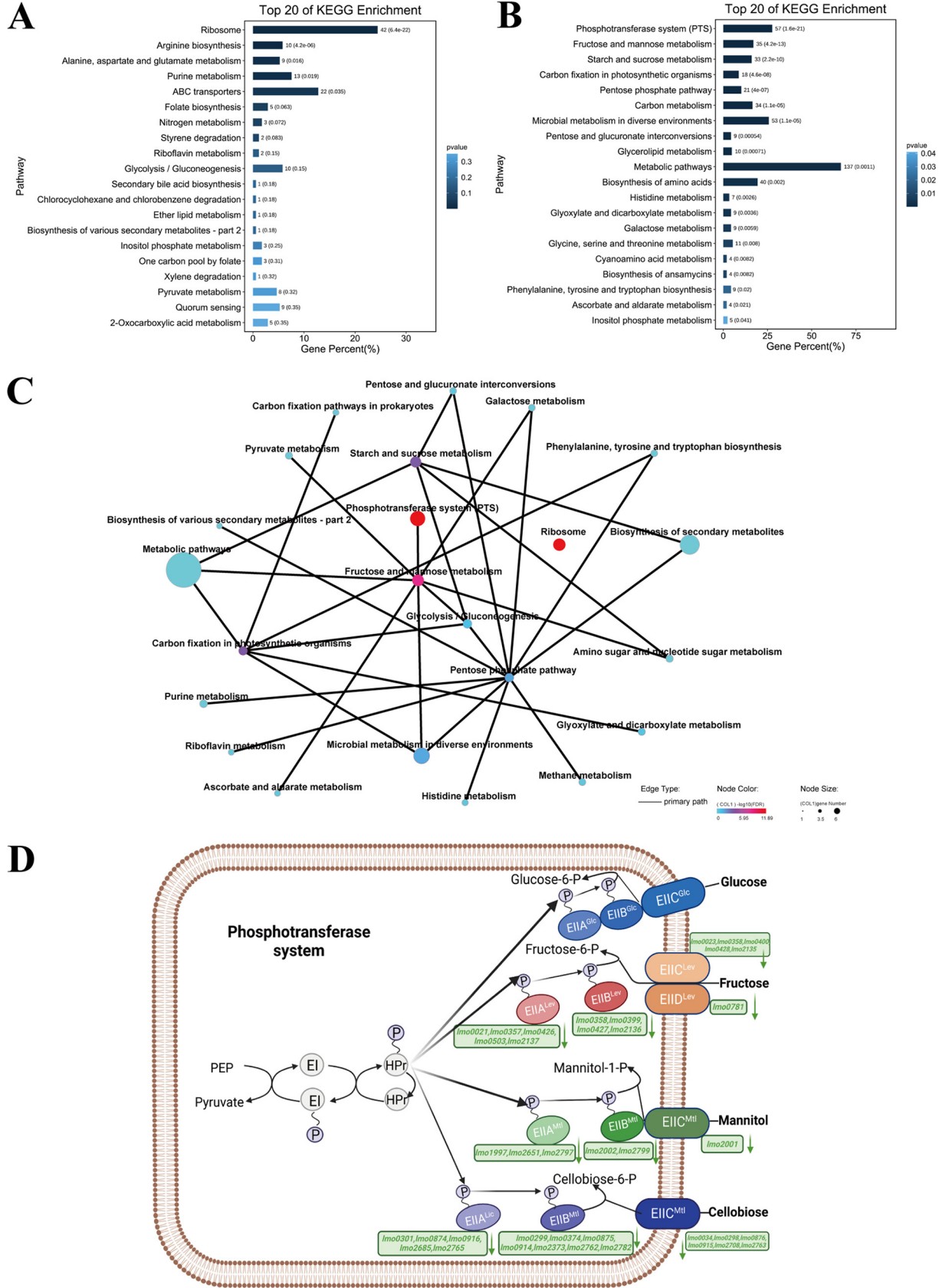

**FIG 5** KEGG pathway analysis of differentially expressed genes (DEGs) in *L. monocytogenes* under LA treatment. (A) Top 20 enriched KEGG pathways for upregulated genes. (B) Top 20 enriched KEGG pathways for downregulated genes. (C) KEGG enrichment network map.

morphologically significant changes under the same concentration of LA treatment also further confirmed the importance of the genes identified in this study for LAR.

Our transcriptomic analysis of *L. monocytogenes* grown in LA revealed pervasive changes in gene expression compared to growth in rich media. In the GO enrichment analysis, "phosphoenolpyruvate-dependent sugar phosphotransferase system," "ribosome," and "protein-N(PI)-phosphohistidine-sugar phosphotransferase activity" were the most significantly enriched subcategories in the biological process, cellular component, and molecular function categories, respectively. PTS mainly phosphorylates various sugars and their derivatives through the phosphorylation cascade and then transports them into the cell (33). It not only participates in carbon and nitrogen central metabolism, regulates iron and potassium homeostasis, and regulates the virulence of certain pathogens, but also mediates stress responses (34). Interestingly, in the KEGG pathway analysis, PTS-related pathways (Fig. 5D) were also significantly enriched. Notably, gene *lmo2248*, identified in Tn-seq experiments, was also shown to be closely related to phosphate transport, indicating that phosphate transport-related genes play an important role in the LAR of *L. monocytogenes*. The results also suggested that ribosome-related genes were significantly upregulated under LA treatment, which was consistent with previous findings (12). Ribosome-related pathways were also found to be significantly enriched under other processing conditions, such as high hydrostatic pressure (35), suggesting that it may be a general stress response.

**Conclusion.** In this study, a genome-wide screening of genetic determinants LAR in *L. monocytogenes* was performed by using Tn-seq, which led to the comprehensive identification of genes that contribute to LAR, including *murF* and *lmo2248*. Moreover, transcriptional analysis revealed that phosphate transport systems were significantly regulated under LA stress conditions, which may play a crucial role in the LAR of *L. monocytogenes*. These findings provide new insights into the molecular mechanisms of LAR in *L. monocytogenes* and may contribute to the development of novel strategies against *L. monocytogenes* and other foodborne pathogens.

## MATERIALS AND METHODS

**Transposon mutant library construction and evaluation.** The high-density transposon mutant library (containing 63,666 unique insertion mutants) was constructed from previous studies in our laboratory (36, 37). Briefly, the temperature-sensitive plasmid pGPA2 was first electroporated into competent cells of *L. monocytogenes*. Then, the plasmid-containing strains were grown overnight at 30°C in BHI supplemented with chloramphenicol, and 200 $\mu$L of this culture was then added to BHI containing gentamicin (25 $\mu$g/mL) and nisin (25 ng/mL) for overnight growth. Next, the cultures were subjected to two consecutive passages in BHI at 37°C to obtain the mutant library. The randomness and coverage of the transposon library were assessed by PCR footprinting and reverse PCR as previously described (38).

**Bacterial strains, mutants, and complement construction.** *L. monocytogenes* EGD-e (ATCC BAA-679, serovar 1/2a) was used throughout this study. The markerless mutant $\Delta lmo2248$ and the complemented strain $\Delta lmo2248 + lmo2248$ were deposited in our laboratory. Markerless gene deletion mutant $\Delta lmo2248$ was created via the Cre-lox recombination system as previously described (38). The primers are listed in Table S1 in the supplemental material. Briefly, the upstream and downstream homology arms of *lmo2248* and the gentamicin maker were PCR amplified with the primers in Table S1. Then, the three DNA fragments were seamlessly cloned (NovoRec plus one-step PCR cloning kit) into the pWS3 vector cleaved with SmaI to obtain the plasmid pWS3-*lmo2248* using EC1000 as the cloning host. The recombinant plasmid was electroporated into *L. monocytogenes* competent cells, and the gentamicin-resistant transformants were cultured at the appropriate temperature and supplemented with the appropriate antibiotics to obtain marked mutants. Next, the plasmid pWS3-erm-cre was electroporated into the marked mutant to obtain the markerless mutants. Plasmids for the in *trans* complementation of the *lmo2248* mutant were produced by PCR amplification of the gene using the primers listed in Table S1. The target gene fragment was then ligated to the downstream region of the PnisA promoter of pMSP3535. Next, the recombinant plasmid was introduced into $\Delta lmo2248$ competent cells and verified by PCR. Unless otherwise mentioned, *L. monocytogenes* and $\Delta lmo2248$ were grown in brain heart infusion (BHI) broth (Hopebio) or agar at 37°C.

**FIG 5** Legend (Continued)

Different nodes represent different KEGG pathways, the size of the node represents the number of genes enriched in the pathway after KEGG enrichment analysis, and the gradient color of the node represents the *P* value of KEGG enrichment analysis. A solid line indicates that there is a connection between a pathway and a pathway or a pathway and a gene, and an isolated node in the figure indicates that the pathway is not directly related to other pathways in the figure. (D) KEGG pathway diagram for the phosphotransferase system (PTS); the downregulated genes are shown in green.

**TABLE 2** Genes related to phosphotransferase system (PTS)[a]

| Synonym | Gene name | Annotation | Fold change | Padj |
|---------|-----------|------------|-------------|------|
| *lmo0021* | *lmo0021* | PTS fructose transporter subunit IIA | 0.2244 | 2.05E-07 |
| *lmo0023* | *lmo0023* | PTS fructose transporter subunit IIC | 0.0331 | 6.73E-13 |
| *lmo0027* | *lmo0027* | PTS beta-glucoside transporter subunit IIABC | 0.0224 | 9.12E-287 |
| *lmo0034* | *lmo0034* | PTS cellobiose transporter subunit IIC | 0.2085 | 2.15E-05 |
| *lmo0298* | *lmo0298* | PTS beta-glucoside transporter subunit IIC | 0.0113 | 2.36E-40 |
| *lmo0299* | *lmo0299* | PTS beta-glucoside transporter subunit IIB | 0.0042 | 7.70E-10 |
| *lmo0301* | *lmo0301* | PTS beta-glucoside transporter subunit IIA | 0.0549 | 2.20E-06 |
| *lmo0357* | *lmo0357* | PTS sugar transporter subunit IIA | 0.3374 | 1.68E-03 |
| *lmo0358* | *lmo0358* | PTS fructose transporter subunit IIBC | 0.2495 | 2.06E-04 |
| *lmo0374* | *lmo0374* | PTS beta-glucoside transporter subunit IIB | 0.2696 | 4.37E-03 |
| *lmo0399* | *lmo0399* | PTS fructose transporter subunit IIB | 0.1191 | 4.36E-10 |
| *lmo0400* | *lmo0400* | PTS fructose transporter subunit IIC | 0.0261 | 8.89E-14 |
| *lmo0426* | *lmo0426* | PTS fructose transporter subunit IIA | 0.2280 | 8.76E-07 |
| *lmo0427* | *lmo0427* | PTS fructose transporter subunit IIB | 0.2581 | 1.04E-03 |
| *lmo0428* | *lmo0428* | PTS fructose transporter subunit IIC | 0.2212 | 5.17E-26 |
| *lmo0503* | *lmo0503* | PTS fructose transporter subunit IIA | 0.0411 | 9.05E-28 |
| *lmo0507* | *lmo0507* | PTS galactitol transporter subunit IIB | 0.1156 | 7.72E-09 |
| *lmo0508* | *lmo0508* | PTS galactitol transporter subunit IIC | 0.1089 | 2.51E-42 |
| *lmo0542* | *lmo0542* | PTS sorbitol transporter subunit IIA | 0.1163 | 2.30E-02 |
| *lmo0543* | *lmo0543* | PTS sorbitol transporter subunit IIBC | 0.1233 | 5.02E-05 |
| *lmo0544* | *lmo0544* | PTS sorbitol transporter subunit IIC | 0.0485 | 9.23E-05 |
| *lmo0738* | *lmo0738* | PTS beta-glucoside transporter subunit IIABC | 0.0009 | 0.00E + 00 |
| *lmo0781* | *lmo0781* | PTS mannose transporter subunit IID | 0.4964 | 3.58E-05 |
| *lmo0874* | *lmo0874* | PTS sugar transporter subunit IIA | 0.1111 | 3.00E-02 |
| *lmo0875* | *lmo0875* | PTS beta-glucoside transporter subunit IIB | 0.0355 | 2.07E-04 |
| *lmo0876* | *lmo0876* | PTS sugar transporter subunit IIC | 0.2438 | 3.55E-04 |
| *lmo0914* | *lmo0914* | PTS sugar transporter subunit IIB | 0.1141 | 5.70E-65 |
| *lmo0915* | *lmo0915* | PTS sugar transporter subunit IIC | 0.0134 | 1.32E-270 |
| *lmo0916* | *lmo0916* | PTS sugar transporter subunit IIA | 0.0051 | 4.39E-70 |
| *lmo1971* | *ulaA* | PTS ascorbate transporter subunit IIC | 0.0581 | 2.09E-07 |
| *lmo1997* | *lmo1997* | PTS mannose transporter subunit IIA | 0.1123 | 5.88E-14 |
| *lmo2000* | *lmo2000* | PTS mannose transporter subunit IID | 0.0514 | 9.43E-34 |
| *lmo2001* | *lmo2001* | PTS mannose transporter subunit IIC | 0.0472 | 1.58E-40 |
| *lmo2002* | *lmo2002* | PTS mannose transporter subunit IIB | 0.1204 | 1.07E-17 |
| *lmo2096* | *lmo2096* | PTS galacticol transporter subunit IIC | 0.1197 | 4.27E-09 |
| *lmo2098* | *lmo2098* | PTS galacticol transporter subunit IIA | 0.1191 | 1.33E-03 |
| *lmo2135* | *lmo2135* | PTS fructose transporter subunit IIC | 0.2246 | 1.88E-04 |
| *lmo2136* | *lmo2136* | PTS fructose transporter subunit IIB | 0.1801 | 8.24E-03 |
| *lmo2137* | *lmo2137* | PTS fructose transporter subunit IIA | 0.0825 | 1.19E-04 |
| *lmo2373* | *lmo2373* | PTS beta-glucoside transporter subunit IIB | 0.4827 | 5.21E-25 |
| *lmo2649* | *ulaA* | PTS system ascorbate transporter subunit IIC | 0.0017 | 0.00E + 00 |
| *lmo2650* | *lmo2650* | MFS transporter | 0.0101 | 6.86E-156 |
| *lmo2651* | *lmo2651* | PTS mannitol transporter subunit IIA | 0.0615 | 2.08E-143 |
| *lmo2665* | *lmo2665* | PTS galacticol transporter subunit IIC | 0.0033 | 0.00E + 00 |
| *lmo2666* | *lmo2666* | PTS galacticol transporter subunit IIB | 0.0026 | 0.00E + 00 |
| *lmo2667* | *lmo2667* | PTS galacticol transporter subunit IIA | 0.0024 | 0.00E + 00 |
| *lmo2685* | *lmo2685* | PTS cellobiose transporter subunit IIA | 0.0828 | 6.78E-50 |
| *lmo2708* | *lmo2708* | PTS cellobiose transporter subunit IIC | 0.4097 | 3.16E-10 |
| *lmo2733* | *lmo2733* | PTS fructose transporter subunit IIABC | 0.4917 | 9.23E-05 |
| *lmo2762* | *lmo2762* | PTS cellobiose transporter subunit IIB | 0.0644 | 3.57E-06 |
| *lmo2763* | *lmo2763* | PTS cellobiose transporter subunit IIC | 0.1640 | 1.13E-16 |
| *lmo2765* | *lmo2765* | PTS cellobiose transporter subunit IIA | 0.3373 | 4.58E-02 |
| *lmo2772* | *lmo2772* | PTS beta-glucoside transporter subunit IIABC | 0.0834 | 1.54E-13 |
| *lmo2782* | *lmo2782* | PTS cellobiose transporter subunit IIB | 0.1655 | 4.48E-02 |
| *lmo2787* | *bvrB* | beta-Glucoside-specific phosphotransferase enzyme II ABC component | 0.2198 | 3.85E-04 |
| *lmo2797* | *lmo2797* | PTS mannitol transporter subunit IIA | 0.0452 | 8.74E-29 |
| *lmo2799* | *lmo2799* | PTS mannitol transporter subunit IIBC | 0.0043 | 5.16E-23 |

[a]*P*adj, adjusted *P* value.

**Antibacterial activity assessment.** To evaluate the antibacterial activity of different concentrations of lactic acid on *L. monocytogenes*, 50 mL of BHI broth with different concentrations of lactic acid (Aladdin, Shanghai, China) (0 [pH 7.03], 0.25 [pH = 6.91], 0.5 [pH = 6.78], 1 [pH = 6.52], 2 [pH = 5.92], and 4 mg/mL [pH = 4.68]) were prepared. Then, the logarithmic phase *L. monocytogenes* was harvested and

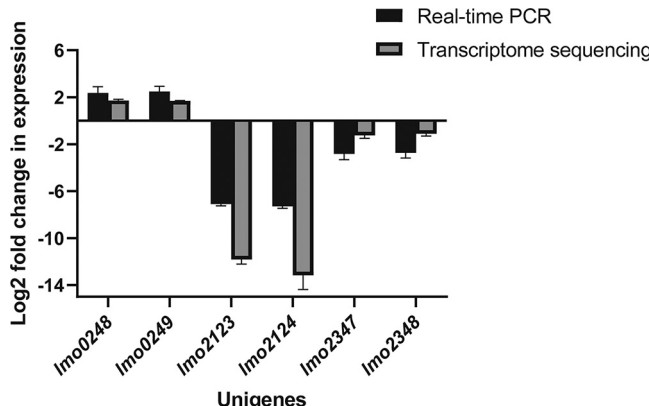

**FIG 6** Real-time qPCR (RT-qPCR) validation of RNA-seq experiments. Gene *drm* was used as the housekeeping control. Each group was performed with 3 biological replicates ($n = 3$).

washed three times (4,000 rpm, 2 min) with phosphate-buffered saline (PBS) and diluted to obtain a final inoculum of $10^6$ CFU/mL as the seed. Next, 1 mL of *L. monocytogenes* ($1.0 \times 10^6$ CFU/mL) seed solution was inoculated into the shake flasks (50 mL BHI broth with different concentrations of lactic acid), and all the flasks were incubated on an orbital shaker (150 rpm at 37°C). Bacterial concentrations were monitored by measuring the optical density at 600 nm ($OD_{600}$) using a UV spectrophotometer (Thermo, USA) after 4 h, 8 h, 12 h, and 24.

**Tn-seq analysis of conditionally essential genes of *L. monocytogenes* under lactic acid treatment.** Tn-seq, a high-throughput tool for the functional genomic study of foodborne pathogens, has previously been reported in detail (21). In this study, we used this technique to perform a genome-wide identification of conditionally essential genes of *L. monocytogenes* under lactic acid treatment. The procedure was similar to that described previously (36). In brief, aliquots containing approximately $10^7$ CFU from the *L. monocytogenes* mutant pool were used to inoculate 100 mL of BHI broth or BHI broth supplemented with 1 mg/mL lactic acid (Aladdin, Shanghai, China). After incubation at 37°C for 16 h, 1 mL culture was used to extract the genomic DNA using the genomic DNA extraction kit (NUOWEIZAN, Nanjing, China). Next, further processing steps (including fragmentation, adapter connection, PCR amplification, etc.) were carried out as previously described (38). Then, high-throughput sequencing was performed on an Illumina HiSeq PE150 instrument (Personalbio, Shanghai, China). This experiment was performed in triplicate.

**Bioinformatics analysis of Tn-seq data.** Tn-seq data analysis was conducted in a previous study (22). After Illumina sequencing, the raw reads were split based on barcodes using Fastp (39), and 16-bp nucleotide fragments corresponding to each reading of the *L. monocytogenes* sequence were mapped to the *L. monocytogenes* genome using Bowtie 2 (40). Then, Integrative Genomics Viewer (IGV) was used to sort and count the results of the alignment. Next, the read counts of each gene were normalized using the following formula: RPTAM (reads per TA sites per million input reads) = (number of reads mapped to a gene $\times 10^6$)/ (total mapped input reads in the sample $\times$ number of TA sites in this gene). Cyber-T (41) (http://cybert.ics .uci.edu/) was used for statistical analysis of RPTAM values between different groups. Genes exhibiting a BH value (Benjamini-Hochberg corrected $P$ value) of <0.05 were determined to be statistically significant.

**Determination of growth curves of the wild type and $\Delta lmo2248$.** A microplate reader (BioTek ELX808IU) was used to determine the effects of lactic acid on bacterial growth. The wild type and the mutant ($\Delta lmo2248$) were grown overnight in BHI. Bacterial cells were inoculated at an initial $OD_{630}$ of 0.005 into 200 $\mu$L BHI and BHI with 1 mg/mL of lactic acid, respectively. The cultures were incubated in the microplate reader at 37°C, and the absorbance at 630 nm ($A_{630}$) was recorded every hour for 22 h. Each experiment was carried out in triplicate.

Overnight cultures of *L. monocytogenes*, $\Delta lmo2248$, and $\Delta lmo2248+ lmo2248$ were diluted 100-fold and inoculated into the fresh BHI broth. Next, 2 mL of the cell cultures (exponential phase) were spun down and resuspended in PBS to an $OD_{600}$ of 1. Then the bacterial solution was serially diluted in a 96-well plate, and 10 $\mu$L was added to the agar plate supplemented with LA (2 mg/mL).

**SEM and TEM.** Bacterial cells (*L. monocytogenes* and $\Delta lmo2248$) were cultured in BHI to the logarithmic growth stage ($OD_{600}$, 0.8) and then treated with 4 mg/mL lactic acid at 37°C for 2 h. Next, the samples were fixed overnight at 4°C in 2.5% glutaraldehyde solution. After a series of pretreatments as previously described (35), the samples were sent to the bio-ultrastructure analysis Lab of Zhejiang University for scanning electron microscopy (GeminiSEM 300, Oberkochen, Germany) and transmission electron microscopy (Hitachi H-7650, Tokyo, Japan) observations.

**Transcriptome profiling.** *L. monocytogenes* was incubated in BHI broth and BHI broth supplemented with 2 mg/mL lactic acid for 16 h. Bacterial cells were centrifuged at 4,000 rpm for 5 min at room temperature and snap-frozen in liquid nitrogen. RNA extraction and quantification were performed as described in previous studies (1, 35). Then, the amplified and purified RNA-seq libraries were sequenced on the NovaSeq 6000 platform (Illumina, San Diego, CA, USA) and 150-bp paired-end reads were generated. Differentially expressed gene (DEG) analysis and enrichment analysis (Gene Ontology

[GO] and Kyoto Encyclopedia of Genes and Genomes [KEGG] enrichment analyses) were performed according to the method described previously (35).

**Validation of transcriptome results by qPCR analysis.** The cDNA was synthesized according to the instructions for the PrimeScript RT reagent kit with gDNA Eraser (TaKaRa, Beijing, China). Then, qPCR was conducted using the TB Green premix *Ex Taq* (Tli RNaseH Plus) kit (Thermo Fisher Scientific, Waltham, MA, USA) and the Applied Biosystems QuantStudio 3 instrument (Thermo Fisher Scientific). Transcript levels, relative to *drm*, of the assayed genes were calculated using QuantStudio Design & Analysis Software 1.3.1 (Thermo Fisher Scientific). Data analysis was performed using the $2^{-\Delta\Delta CT}$ method. Each group contained 3 biological replicates.

**Statistical analysis.** All analyses of significance were performed using one-way analysis of variance (ANOVA) and Duncan's multiple-range test (DMRT). In all analyses, only when *P* was <0.05, were data considered statistically significant. All tests were conducted in triplicate.

**Data availability.** The raw RNA-seq data are available at the NCBI Sequence Read Archive (SRA) under BioProject accession no. PRJNA862260, and the Tn-seq data are available at NCBI SRA under BioProject accession no. PRJNA862615.

## SUPPLEMENTAL MATERIAL

Supplemental material is available online only.
**SUPPLEMENTAL FILE 1**, PDF file, 0.1 MB.

## ACKNOWLEDGMENTS

This work was supported by the Natural Science Foundation of Shandong Province, China (ZR2019ZD21), the Taishan Scholars Program of Shandong Province, China (ts20190955), the Project of Shandong Province Higher Educational Outstanding Youth Innovation Team (2019KJF011), and the National Key Research and Development Program of China (2019YFE0103900).

We declare that we have no conflict of interest.

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
