## [Reviewer comments · Microbiology Spectrum]

Microbiology Spectrum

New insights into the lactic acid resistance determinants of *Listeria monocytogenes* based on Tn-seq and RNA-Seq analyses

Xiayu Liu, Xinxin Pang, Yansha Wu, Yajing Wu, Linan Xu, Qihe Chen, Jianrui Niu, and Xinglin Zhang

Corresponding Author(s): Xinglin Zhang, Linyi University

Review Timeline:

Submission Date:	July 18, 2022
Editorial Decision:	October 26, 2022
Revision Received:	November 8, 2022
Accepted:	November 15, 2022

Editor: Luxin Wang

Reviewer(s): Disclosure of reviewer identity is with reference to reviewer comments included in decision letter(s). The following individuals involved in review of your submission have agreed to reveal their identity: Augustine Kwaku Agyekum (Reviewer #1)

Transaction Report:

DOI: <https://doi.org/10.1128/spectrum.02750-22>

October 26, 2022

Prof. Xinglin Zhang
Zhejiang University
Yuhangtang Rd.866
hangzhou
China

Re: Spectrum02750-22 (New insights into the lactic acid resistance determinants of *Listeria monocytogenes* based on Tn-seq and RNA-Seq analyses)

Dear Prof. Xinglin Zhang:

Thank you for submitting your manuscript to Microbiology Spectrum. The reviewers have submitted their comments and recommend "modification". Their comments are attached with this email.

Link Not Available

Sincerely,

Luxin Wang

Journals Department
Reviewer comments:

Reviewer #3 (Public repository details (Required)):

Tn-seq and RNA-seq data should be deposited in a sequence data archive. Perhaps I have missed it, but I did not see accession numbers in the manuscript.

Reviewer #3 (Comments for the Author):

This is a concise manuscript that studies the lactic acid tolerance response of the foodborne pathogen *Listeria monocytogenes* through transposon mutant library profiling (Tn-seq) and genome-wide gene expression analysis (RNA-seq). While technically correct, the manuscript may need to be improved further as described below:

Major points:

The introduction should be adapted to include more text on our current knowledge of lactic acid tolerance mechanism in *L. monocytogenes* and previous studies that have identified loci in the *Listeria monocytogenes* genome that contribute to this response, e.g. Begley et al., 2003, FEMS Microbiol Lett 218:31-38, Madeo et al., 2012, FEMS Microbiol Lett 326:137-43)

The pH of the medium with the different concentrations of lactic acid should be measured and reported in the manuscript. (L. 94 - 101)

L. 241. Significantly more detail should be provided on how the mutant and the complemented strain were generated. In addition, more detail could be provided to describe how the transposon mutant library was generated and validated (l. 263).

Sequence data needs to be uploaded to a data repository (e.g. the Short Read Archive)

Minor points:

While I had no problems understanding the manuscript, there are some minor grammatical or textual issues throughout which could benefit from corrections.

L. 58: all use of antibiotics will select for resistance, not just abuse.

L. 67-73: it is better to give a specific, relevant examples rather than adding 'etc' as that is not particularly informative.

L. 115 and L. 198 STRING analysis shows whether genes are interacting with each other, not necessarily whether they are related to each other as that suggests they are homologs, which I believe is not what the others mean to say. Have homologs of this gene been characterised in other Gram-positive bacteria (e.g. *Bacillus subtilis*, *Lactobacillus* or *Lactococcus*)?

L. 121 It is confusing that murF is apparently an essential gene, but the authors do identify a transposon mutant in that gene. How can this be explained?

L. 124 Complementary should be complemented

L. 155. This is the first time the abbreviation DEGs is used and it should be explained here.

In the section on the RNA-seq analysis, it should be mentioned whether lmo2248 was significantly up- or down-regulated to connect these analyses with the Tn-seq data presented previously in the manuscript.

Staff Comments:

Preparing Revision Guidelines

Please return the manuscript within 60 days; if you cannot complete the modification within this time period, please contact me. If you do not wish to modify the manuscript and prefer to submit it to another journal, please notify me of your decision

immediately so that the manuscript may be formally withdrawn from consideration by Microbiology Spectrum.

Response to reviewers' comments

Dear Prof Luxin Wang,

On behalf of my co-authors, we thank you very much for giving us an opportunity to revise our manuscript, we appreciate editor and reviewers very much for their positive and constructive comments and suggestions on our manuscript entitled "New insights into the lactic acid resistance determinants of *Listeria monocytogenes* based on Tn-seq and RNA-Seq analyses".

We have studied the comments of editor and reviewers carefully and have made revision which marked in red in the revised manuscript. We have tried our best to revise this manuscript according to the comments. Attached please find the revised version, which we would like to submit for your kind consideration. And responses to reviewers' comments are appended below.

We would like to express our great appreciation to you and reviewers for comments on our paper. Looking forward to hearing from you.

Yours sincerely,

Dr. Xinglin Zhang
College of Agriculture and Forestry
Linyi University
Linyi 276000 P.R.China
Tel: +86-571-86984316
E-Mail: zhangxinglin@lyu.edu.cn

List of Responses.

Reviewer #3: This is a concise manuscript that studies the lactic acid tolerance response of the foodborne pathogen *Listeria monocytogenes* through transposon mutant library profiling (Tn-seq) and genome-wide gene expression analysis (RNA-seq). While technically correct, the manuscript may need to be improved further as described below.

Reply: We thank the reviewer for his/her compliments on our manuscript. Below we will reply in a point-by-point fashion to the reviewer's comments and indicate the alterations we have made to the manuscript.

Major points:

1. Tn-seq and RNA-seq data should be deposited in a sequence data archive. Perhaps I have missed it, but I did not see accession numbers in the manuscript.

Reply: We appreciate for reviewer's advice and we have added the accession numbers in the manuscript (line 359-362). The raw RNA-seq data are available at NCBI Sequence Read Archive (NCBI SRA) under BioProject accession no. PRJNA862260 and the Tn-seq data are available at NCBI SRA under BioProject accession no. PRJNA862615.

2. The introduction should be adapted to include more text on our current knowledge of lactic acid tolerance mechanism in *L. monocytogenes* and previous studies that have identified loci in the *Listeria monocytogenes* genome that contribute to this response, e.g. Begley et al., 2003, FEMS Microbiol Lett 218:31-38, Madeo et al., 2012, FEMS Microbiol Lett 326:137-43)

Reply: We appreciate the suggestion very much. We have made relevant supplements in the Introduction and added more references. (line 77-85)

3. The pH of the medium with the different concentrations of lactic acid should be measured and reported in the manuscript. (L. 94 - 101)

Reply: We appreciate for reviewer's advice and we have supplemented the data with pH values of the medium at different lactic acid concentrations. (line 106-112) (line 277-280)

4. L. 241. Significantly more detail should be provided on how the mutant and the complemented strain were generated. In addition, more detail could be provided to describe how the transposon mutant library was generated and validated (l. 263).

Reply: We appreciate for reviewer's constructive advice and we have added a detailed description of how the mutant strains, complementary strains and mutant libraries are generated and the relevant references were cited. (line 246-255) (line 259-274)

5. Sequence data needs to be uploaded to a data repository (e.g. the Short Read Archive).

Reply: We appreciate for reviewer's suggestion and we have added the accession numbers in the manuscript (line 359-362). The raw RNA-seq data are available at NCBI Sequence Read Archive (NCBI SRA) under BioProject accession no. PRJNA862260 and the Tn-seq data are available at NCBI SRA under BioProject accession no. PRJNA862615.

Minor points:

6. L. 58: all use of antibiotics will select for resistance, not just abuse.

Reply: We thank the reviewer for carefully reading and pointing out the deficiencies of our manuscript. We have revised this section. (line 57-58)

7. L. 67-73: it is better to give a specific, relevant examples rather than adding 'etc' as that is not particularly informative.

Reply: We appreciate for reviewer's advice and we have supplemented relevant examples and specific instructions. (line 63-67, 69-75)

8. L. 115 and L. 198 STRING analysis shows whether genes are interacting with each other, not necessarily whether they are related to each other as that suggests they are homologs, which I believe is not what the others mean to say. Have homologs of this gene been characterised in other Gram-positive bacteria (e.g. *Bacillus subtilis*, *Lactobacillus* or *Lactococcus*)?

Reply: Thanks for the reviewer's constructive remarks. We have removed these inappropriate descriptions in the manuscript. Although we did not find a homologue of *Imo2248* at NCBI, a description of its downstream gene *Imo2249* (encoding a low-affinity inorganic phosphate transporter) was added in the manuscript. (line 204-206).

9. L. 121 It is confusing that *murF* is apparently an essential gene, but the authors do identify a transposon mutant in that gene. How can this be explained?

Reply: The reviewer raises an excellent point. In this study, transposon insertion mutation of *murF* was indeed identified, but due to its low number of reads, it indicates that this mutant strain is also sick and grows slowly, and if it is passed on for more generations it may be competed out and become essential. Because essential genes are indispensable genes for organisms to grow and reproduce offspring under certain environment. However, being essential is highly dependent on the circumstances in which an organism lives.

Synonym	Total TAs	C1	C2	C3	L1	L2	L3	BHI_count	Gene_size	RefSeq	product	P-value	FC(BHI/L)	BH
Imo0856	68	230	362	163	31	9	17	755	1374	murF	alanyl-D- glutamyl-2,6-diamino pimelate-	0.035599757	13.524201	0.006643

10. L. 124 Complementary should be complemented.

Reply: We appreciate the reviewer for this suggestion. Following the reviewer's suggestion, we have made changes to it. (line 131)

11. L. 155. This is the first time the abbreviation DEGs is used and it should be explained here.

Reply: We thank the reviewer for pointing this out. We have replaced the DEGs with “differentially expressed genes (DEGs).” (line 157-158)

12. In the section on the RNA-seq analysis, it should be mentioned whether *Imo2248* was significantly up- or down-regulated to connect these analyses with the Tn-seq data presented previously in the manuscript.

Reply: We appreciate the reviewer for the valuable suggestion. We have supplemented the description of changes in *Imo2248* at the transcriptional level in the Results (line 159-160) and Discussion (line 208-210) sections.

November 15, 2022

Prof. Xinglin Zhang
Linyi University
linyi
China

Re: Spectrum02750-22R1 (New insights into the lactic acid resistance determinants of *Listeria monocytogenes* based on Tn-seq and RNA-Seq analyses)

Dear Prof. Xinglin Zhang:

I am pleased to inform you that your manuscript has been accepted, and I am forwarding it to the ASM Journals Department for publication. You will be notified when your proofs are ready to be viewed.

Sincerely,

Luxin Wang
Editor, Microbiology Spectrum
